# Clinical Significance of Lymph-Node Ratio in Determining Supraclavicular Lymph-Node Radiation Therapy in pN1 Breast Cancer Patients Who Received Breast-Conserving Treatment (KROG 14-18): A Multicenter Study

**DOI:** 10.3390/cancers11050680

**Published:** 2019-05-16

**Authors:** Jaeho Kim, Won Park, Jin Hee Kim, Doo Ho Choi, Yeon-Joo Kim, Eun Sook Lee, Kyung Hwan Shin, Jin Ho Kim, Kyubo Kim, Yong Bae Kim, Sung-Ja Ahn, Jong Hoon Lee, Mison Chun, Hyung-Sik Lee, Jung Soo Kim, Jihye Cha

**Affiliations:** 1Department of Radiation Oncology, Dongsan Medical Center, Keimyung University School of Medicine, Daegu 42601, Korea; rainfield@naver.com; 2Department of Radiation Oncology, Samsung Medical Center, Sungkyunkwan University School of Medicine, Seoul 06351, Korea; wonro.park@samsung.com (W.P.); dohochoi@hanmail.net (D.H.C.); 3Center for Breast Cancer, Research Institute and Hospital, National Cancer Center, Goyang 10408, Korea; jane2000md@gmail.com (Y.-J.K.); eslee@ncc.re.kr (E.S.L.); 4Department of Radiation Oncology, Seoul National University College of Medicine, Seoul 03080, Korea; radiat@snu.ac.kr (K.H.S.); jinho.kim.md@gmail.com (J.H.K.); 5Department of Radiation Oncology, Ewha Womans University Mokdong Hospital, Ewha Womans University School of Medicine, Seoul 07804, Korea; kyubokim.ro@gmail.com; 6Department of Radiation Oncology, Yonsei Cancer Center, Yonsei University College of Medicine, Seoul 03722, Korea; ybkim3@yuhs.ac; 7Department of Radiation Oncology, Chonnam National University Medical School, Gwangju 61469, Korea; ahnsja@chonnam.ac.kr; 8Department of Radiation Oncology, St. Vincent′s Hospital, The Catholic University of Korea College of Medicine, Seoul 06591, Korea; koppul@catholic.ac.kr; 9Department of Radiation Oncology, Ajou University School of Medicine, Suwon 16499, Korea; chunm@ajou.ac.kr; 10Department of Radiation Oncology, Dong-A University Hospital, Dong-A University School of Medicine, Busan 49201, Korea; hyslee@dau.ac.kr; 11Department of Radiation Oncology, Chonbuk National University Medical School, Jeonju 54907, Korea; jskim@chonbuk.ac.kr; 12Department of Radiation Oncology, Wonju Severance Christian Hospital, Wonju 26426, Korea; lukanus@yonsei.ac.kr

**Keywords:** breast cancer, radiotherapy, lymph-node ratio, disease-free survival

## Abstract

This study evaluated the clinical significance of the lymph-node ratio (LNR) and its usefulness as an indicator of supraclavicular lymph-node radiation therapy (SCNRT) in pN1 breast cancer patients with disease-free survival (DFS) outcomes. We retrospectively analyzed the clinical data of patients with pN1 breast cancer who underwent partial mastectomy and taxane-based sequential adjuvant chemotherapy with postoperative radiation therapy in 12 hospitals (*n* = 1121). We compared their DFS according to LNR, with a cut-off value of 0.10. The median follow-up period was 66 months (range, 3–112). Treatment failed in 73 patients (6.5%) and there was no significant difference in DFS between the SCNRT group and non-SCNRT group. High LNR (>0.10) showed significantly worse DFS in both univariate and multivariate analyses (0.010 and 0.033, respectively). In a subgroup analysis, the effect of SCNRT on DFS differed significantly among patients with LNR > 0.10 (*p* = 0.013). High LNR can be used as an independent prognostic factor for pN1 breast cancer patients treated with partial mastectomy and postoperative radiotherapy. It may also be useful in deciding whether to perform SCNRT to improve DFS.

## 1. Introduction

In addition to surgery, chemotherapy and radiotherapy are both important in curative breast cancer treatment. Radiotherapy has a significant role in the removal of microscopic tumor cells from remnant breast tissue after breast-conserving surgery [1]. Radiotherapy is used to treat not only the remaining breast tissue, but also the tumor cells in the regional lymphatic system, including the axillar and internal mammary lymph nodes. This prevents locoregional failure in patients with breast-conserving surgery and lymph-node metastasis after axillary-lymph-node dissection. It can also reduce distant metastasis [2]. The Danish Breast Cancer Cooperative Group reported the benefits of radiation therapy targeting the supraclavicular area in high-risk breast cancer patients [3,4]. The current guidelines generally recommend that elective nodal irradiation (ENI) be applied to the regional lymphatics as well as the whole breast in locally advanced breast cancer [5,6,7].

The use of ENI in pathological N1 breast cancer patients is still controversial. Taxane-based chemotherapy has been used as an effective adjuvant therapy for breast cancer for decades, reducing the importance of elective nodal irradiation in low-risk breast cancer [8,9,10]. There is no consensus on whether ENI should be administered to patients with a low risk of regional lymphatic metastasis, such as those with N1 breast cancer [6,11,12,13]. The current National Comprehensive Cancer Network (NCCN) guidelines also recommend that ENI be strongly considered in N1 breast cancer [14].

In a study based on existing Korean Radiation Oncology Group (KROG) 14-18 data, supraclavicular lymph-node radiation therapy (SCNRT) was ineffective in N1 breast cancer patients undergoing taxane-based chemotherapy [15]. However, several studies have reported that various risk factors affect the outcomes of patients with N1 breast cancer, and claimed that ENI can be beneficial in patients with N1 breast cancer, depending on the risk factors present [16,17,18]. The lymph-node ratio (LNR), defined as the proportion of positive axillary lymph nodes among the total number of axillary lymph nodes removed, is recognized as one of these risk factors. Although the absolute number of axillary lymph nodes affected by metastasis is associated with a poor prognosis in the American Joint Committee on Cancer (AJCC) guidelines, axillary-lymph-node dissection techniques may differ across different institutions [19]. Therefore, the number of lymph nodes removed may differ, even when patients have the same numbers of metastatic lymph nodes [20]. Several studies have investigated whether LNR can be used to ensure more accurate nodal staging [19,20,21]. In a previous single-center retrospective study that examined the relationship between LNR and SCNRT, LNR was found to have utility as an indicator of the suitability of SCNRT [22]. Against this background, it was investigated whether LNR can be used as an index of the suitability of SCNRT in multicenter studies.

## 2. Materials and Methods

### 2.1. Study Design and Patients

In this study, the records of patients diagnosed with N1 breast cancer who underwent breast-conserving surgery between January 2006 and December 2010 at one of 12 hospitals that are members of KROG were examined. Patients who underwent Adriamycin/Taxol (AT) chemotherapy and post-lumpectomy radiotherapy for N1 breast cancer within this period were included in this study. The eligibility criteria were patients with N1 breast cancer who underwent breast-conserving surgery and axillary-lymph-node dissection, who completed postoperative AT chemotherapy and radiotherapy as planned, and for whom information regarding the pathological features of the tumor was available. The exclusion criteria were patients who received neoadjuvant chemotherapy or chemotherapy other than AT, had a previous history of malignancy, or were male. Patients with fewer than 10 dissected lymph nodes in total were also excluded from the study to ensure accurate lymph node evaluation. The Institutional Review Board of each participating hospital approved the current study.

The patient data collected were age; the pathological features of each tumor, such as the tumor size, number of tumors, and resection margin; number of positive lymph nodes; histological grade; presence of lymphovascular invasion; and expression status of the estrogen receptor (ER), progesterone receptor (PR), and human epidermal growth factor receptor 2 (HER2). Positivity for ER or PR was defined as an Allred score of 3–8 on immunohistochemistry (IHC). HER2 positivity was defined as either 3+ on IHC staining or 2+ on IHC with a positive fluorescence in situ hybridization or chromogenic in situ hybridization signal. The molecular subtype of each breast cancer was categorized as follows: ER+ or PR+, and HER2− (luminal A); ER+ or PR+, and HER2+ (luminal B); ER−, PR−, and HER2+ (HER2 enriched); or ER−, PR−, and HER2− (triple negative). The optimal cut-off value for LNR was determined with an analysis of the area under the curve (AUC) of a receiver-operating characteristic (ROC) curve. A value of 0.10, for which the sensitivity and specificity were highest, was chosen as the optimal cut-off point for LNR.

### 2.2. Treatments

In this study, patients who had undergone more than 10 lymph-node dissections after axillary-lymph-node dissection were evaluated. The patients were treated with adjuvant chemotherapy consisting of doxorubicin and cyclophosphamide, followed by paclitaxel or docetaxel, and then by the appropriate adjuvant hormone therapy based on the presence of hormone receptors and the HER2 receptor. All patients underwent whole-breast irradiation (WBI) and tumor-bed boost, and SCNRT was selected according to each institution’s policy or physician’s preference. 

The dose of irradiation for the whole breast was 45.0–60.4 Gy at 1.8–3.0 Gy per fraction and the dose for the tumor bed was 4.0–19.8 Gy at 1.8–4.0 Gy per fraction. The radiation dose to the supraclavicular lymph nodes (SCN) was 45.0–50.4 Gy at 1.8–2.0 Gy per fraction. The borders of each field of WBI or WBI+SCNRT were defined differently in the 12 hospitals according to each institutional policy. The axillary lymph nodes were not intentionally irradiated, but level I and a proportion of level II axillary lymph nodes were covered during WBI while a proportion of the level II and level III axillary lymph nodes and the SCN were irradiated during SCNRT. The inclusion of the internal mammary lymph node in the radiation field was determined according to each hospital′s policy, considering the location of the primary tumor, pathologic findings, and status of the metastatic lymph nodes.

### 2.3. Follow-Up and Endpoints

Disease-free survival (DFS) was defined as the time from the date of diagnosis to any recurrence. The patients were followed up on every 3–6 months after surgery with history and physical examinations in each hospital. Mammography was performed every 12 months. Additional imaging studies were performed in patients with suspicious clinical signs or symptoms.

### 2.4. Statistical Analysis

Data were analyzed with SPSS ver. 20.0 for Windows (SPSS Inc., Chicago, IL, USA). The Kaplan–Meier method was used to analyze disease-free survival (DFS) and statistical significance was determined with a log-rank test. Cox’s stepwise regression analysis was used for the multivariable analysis. Statistically significant variables in the univariate analysis (*p* < 0.05) were included in the Cox’s regression model. Statistical significance was set at *p* < 0.05. 

## 3. Results

The characteristics of the patients with pN1 breast cancer are summarized in Table 1. In total, 1121 patients satisfied the inclusion criteria for this study and were enrolled. Among them, 745 patients did not undergo SCNRT and 376 patients did undergo SCNRT. The presence of an extensive intraductal component (EIC), lymphovascular invasion (LVI), number of positive lymph nodes, LNR, and the presence of an extracapsular extension (ECE) occurred significantly more frequently in the SCNRT group than in the non-SCNRT group. LNR was statistically independent of other prognostic factors, such as age, type of surgery, T stage, resection margin, LVI, molecular subtype, and histological grade.

The median follow-up time was 66 months (range, 3–112). The overall 5-year DFS was 93.7%. The 5-year DFS in the subgroup was 92.8% in the SCNRT group and 94.1% in the non-SCNRT group. The patterns of failure are shown in Table 2. Distant metastasis was the major pattern of failure, and regional recurrence limited to the SCN occurred in less than 1% of the total patients. 

A univariate analysis of DFS showed that T stage, LVI, histological grade, luminal type, and LNR were significant factors affecting DFS. In a multivariate analysis of these factors, T stage, LVI, histological grade, and LNR significantly affected DFS (Table 3 and Table 4).

A subgroup analysis according to SCNRT was performed to analyze the risk of recurrence according to differences in LNR. In this analysis, the risk of recurrence differed significantly according to LNR in the non-SCNRT group, but there was no such difference in the SCNRT group (Figure 1). The use of SCNRT reduced the difference in the incidence of recurrence according to LNR.

## 4. Discussion

Although many studies have reported that ENI is not required by patients with pN0 breast cancer, it is widely accepted that postoperative locoregional radiation therapy reduces locoregional recurrence and mortality in patients with lymph-node-positive breast cancer [23]. However, its effectiveness in N1 breast cancer patients treated with systemic chemotherapy is still unclear [24,25]. In this context, the NCCN guidelines recommend the use of SCNRT in patients with N1 breast cancer as for level of evidence IIB [14].

Appropriate chemotherapy for breast cancer patients not only prevents systemic metastasis of the cancer, but also reduces the risk of locoregional recurrence [26]. The recurrence rate after treatment has been steadily declining with the development of surgical and adjunct therapies, and questions about the utility of SCNRT for early breast cancer patients have begun to emerge [27]. The frequency of adverse effects, such as lymphedema, increases when SCNRT is used with systemic chemotherapy. A previous study based on the KROG 14–18 patient data used in the present study showed that SCNRT was unnecessary in the N1 patient group and increased adverse effects [15,28]. In the case of another side effect, brachial plexopathy, the risk showed 1.3% in the conventional 50 Gy SCNRT group, but the risk was increased with adjuvant chemotherapy or total dose over 50 Gy to the brachial plexus [29].

Although the risk is low, treatment failure in N1 breast cancer occurs in regional and distant areas, and may be related to microscopic tumor cells in the regional lymphatic system, which can be removed with SCNRT. SCNRT still plays an important role in breast cancer treatment within this context. This may justify the use of SCNRT in selected patients with high-risk N1 breast cancer, rather than in all patient groups. Using SCNRT for N1 breast cancer is an interesting issue in the field of radiation oncology, and several studies have recommended that SCNRT be performed after various risk factors are considered, but not in all patient groups [17,18]. SCNRT can entail radiation side effects, such as lymphedema, brachial plexopathy, and radiation pneumonitis, so it should not be performed for small potential benefits alone. The benefits of SCNRT and its adverse effects in N1 breast cancer patients are highly controversial, and are affected by other modalities, such as the chemotherapy regimen and tumor characteristics. Therefore, its benefits should be clarified in a prospective study.

In this study, the utility of LNR was investigated as a factor to be considered in selecting SCNRT as a treatment for N1 breast cancer patients. Several studies have reported that nodal staging using LNR can predict the prognosis more effectively than conventional AJCC nodal staging [19,20,30]. In patients with N1 breast cancer with a relatively small number of metastatic lymph nodes, the prognosis can vary according to the total number of dissected lymph nodes, even in patients with the same number of metastatic lymph nodes. Therefore, this study investigated whether the utility of SCNRT can be estimated from LNR. This study was conducted with KROG 14–18 patients, and previous studies using the same patient group showed that SCNRT was ineffective in N1 breast cancer patients [15]. In the present study, it was demonstrated that SCNRT may be beneficial in patients with high LNRs, despite the use of taxane-based chemotherapy.

This study had several limitations. In addition to the limitations of all retrospective studies, there was no definitive standard according to which all institutions decided whether or not to administer SCNRT. Similarly, internal mammary node irradiation was also applied without clear criteria, which blurred the conclusions of this study. The difficulty in evaluating adverse effects was another limitation of this study. It is well known that SCNRT induces lymphedema. According to preliminary findings reported by Coen et al., the addition of regional lymph-node radiotherapy significantly increased the risk of lymphedema from 1.8% to 8.9% (*p* = 0.0001) [28]. When considering the use of SCNRT, clinicians should weigh the potential benefits of SCNRT in disease control against the increased risk of lymphedema. A previous study based on existing KROG 14–18 data showed that lymphedema occurred in 16.6% of patients after WBI+ and SCNRT, but in only 10.7% of patients after WBI alone (*p* = 0.04) [15]. To compare the benefits of SCNRT, a prospective study is required to establish more precise criteria. In this study, regional recurrence in the SCN, which is the main target of SCNRT, occurred in only 1% of patients. This is a very low recurrence rate upon which to base the claim that SCNRT should be performed in all patients. It is difficult to accept that SCNRT should be used to control recurrence, which occurs in only 1% of patients, given its adverse effects mentioned above. This study has shown that SCNRT also significantly reduced the risk of distant metastasis, as well as regional lymph-node recurrence. This may be because SCNRT controls the tumor burden in the SCN region, which may be the seed bed of distant metastases. Distant metastasis is the main failure pattern in N1 breast cancer patients, so this study has demonstrated that SCNRT has some therapeutic benefits over and above the control of regional recurrence. Lastly, the fact that there are characteristic differences between SCNRT group and non-SCNRT group is a weakness of the study. These characteristics are significantly higher in the SCNRT group and are generally known as poor prognosis factors, such as LVI, LNR, and ECE [13]. Despite these poor prognosis factors in the SCNRT group, there was no difference in recurrence rate between the two groups and it can be interpreted as indicating the effect of SCNRT.

Despite the several limitations described above, this multicenter study suggests that the use of SCNRT in patients receiving taxane-based chemotherapy may be beneficial if SCNRT is performed selectively. The study is important because it offers another direction of selective SCNRT for N1 breast cancer. Selective SCNRT can reduce unnecessary radiation exposure in N1 breast cancer patients and reduce the recurrence rate in appropriate patients, with a more patient-specific treatment.

## 5. Conclusions

In this DFS analysis, the patients with high LNR (>0.10) showed benefit with DFS outcomes by SCNRT. This study has shown that LNR can be used as an independent prognostic factor in patients with N1 breast cancer and as a useful index when determining whether to perform SCNRT. However, we did not establish whether the benefits of this treatment are sufficient to risk the associated adverse effects. Prospective studies are required to examine these issues.

## Figures and Tables

**Figure 1 cancers-11-00680-f001:**
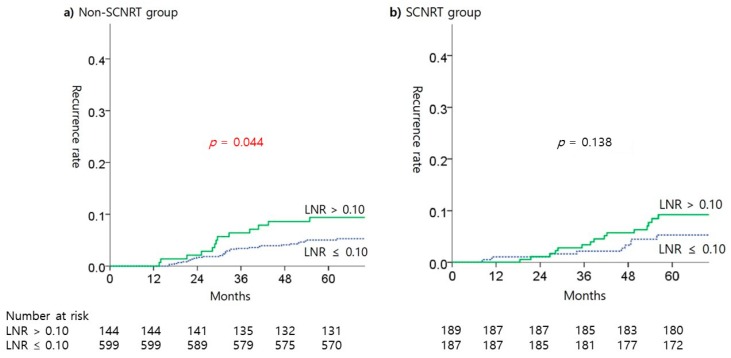
Kaplan-Meier estimates of recurrent rate according to LNR in SCNRT treatment subgroups.

**Table 1 cancers-11-00680-t001:** Patient characteristics.

Characteristics		Number of Patients (%)		*p* Value
		Non-SCNRT	SCNRT	
		(*n* = 745)	(*n* = 376)	
Age (years)	≤45	442 (59.3)	221 (58.8)	0.859
	>45	303 (40.7)	155 (41.2)	
OP site	Left	343 (46.0)	193 (51.3)	0.094
	Right	402 (54.0)	183 (48.7)	
Pathology	IDC	702 (94.2)	346 (92.0)	0.157
	Others	43 (5.8)	30 (8.0)	
T stage	T1	388 (52.1)	181 (48.1)	0.282
	T2	351 (47.1)	192 (51.1)	
	T3	6 (0.8)	2 (0.5)	
	T4	0 (0.0)	1 (0.3)	
Number of	Single	615 (82.6)	318 (84.6)	0.392
tumors	Multiple	130 (17.4)	58 (15.4)	
Resection	Clear	682 (92.2)	351 (93.9)	0.588
margin	Less than 1 mm	51 (6.9)	20 (5.3)	
	Positive	7 (0.9)	3 (0.8)	
	Unknown	5	2	
EIC	(−)	344 (71.2)	225 (63.9)	0.025
	(+)	139 (28.8)	127 (36.1)	
	Unknown	262	24	
LVI	(−)	360 (49.0)	81 (24.0)	<0.001
	(+)	375 (51.0)	256 (76.0)	
	Unknown	10	39	
HG	I or II	452 (61.9)	235 (63.0)	0.725
	III	278 (38.1)	138 (37.0)	
	Unknown	15	3	
Anti-HER2	(−)	678 (91.0)	339 (90.2)	0.644
therapy	(+)	67 (9.0)	37 (9.8)	
Dissected	<20	456 (61.2)	245 (65.2)	0.197
LNs	≥20	289 (38.8)	131 (34.8)	
Number of	1	509 (68.5)	121 (32.2)	<0.001
positive LNs	2	161 (21.7)	145 (38.6)	
	3	73 (9.8)	110 (29.3)	
	Unknown	2	0	
LNR	≤0.10	599 (80.6)	189 (50.3)	<0.001
	>0.10	144 (19.4)	187 (49.7)	
	Unknown	2	0	
ECE	(−)	369 (52.4)	92 (31.8)	<0.001
	(+)	335 (47.6)	197 (68.2)	
	Unknown	41	87	
Hormone	(−)	172 (23.1)	86 (22.9)	0.917
therapy	(+)	571 (76.9)	290 (77.1)	
	Unknown	2	0	
Molecular	Luminal A	498 (67.0)	247 (65.7)	0.19
subtype	Luminal B	71 (9.5)	49 (13.0)	
	HER2-enriched	53 (7.1)	17 (4.5)	
	Triple negative	122 (16.4)	63 (16.8)	
	Unknown	1	0	

Abbreviations: SCNRT, supraclavicular lymph node radiation therapy; OP, operation; IDC, invasive ductal carcinoma; EIC, extensive intraductal component; LVI, lymphovascular invasion; HG, histologic grade; HER2, human epidermal growth factor receptor 2; LNR, lymph-node ratio; ECE, extracapsular extension.

**Table 2 cancers-11-00680-t002:** Patterns of failure.

Outcome	No. Patients (%)
Follow-up (months)	
Median (range)	66 (3–112)
Patterns of failure	
NED	1048 (93.5)
LR only	8 (0.7)
RR only	5 (0.4)
DM only	45 (4.0)
LR + DM	1 (0.1)
RR + DM	11 (1.0)
LR + RR + DM	3 (0.3)

Abbreviations: NED, no evidence of disease; LR, local recurrence; RR, regional recurrence; DM, distant metastasis.

**Table 3 cancers-11-00680-t003:** Univariate analysis of disease-free survival.

Characteristics		No. (%)	5-Year DFS	*p* Value
T stage	T1	569 (50.7)	97.1	<0.001
	T2–4	552 (49.2)	90.2	
Number of	Single	933 (83.2)	93.9	0.953
tumors	Multiple	188 (16.8)	93.1	
Resection	≥1 mm	1033 (92.1)	93.8	0.634
margin	<1 mm	81 (7.2)	92.5	
EIC	(−)	569 (50.8)	94.8	0.213
	(+)	266 (23.7)	91.8	
LVI	(−)	441 (39.3)	97.0	0.001
	(+)	631 (56.3)	91.4	
HG	I or II	687 (61.3)	96.1	<0.001
	III	416 (37.1)	89.9	
LNR	≤0.10	788 (70.3)	94.9	0.010
	>0.10	331 (29.5)	90.8	
ECE	(−)	461 (41.1)	92.7	0.110
	(+)	532 (47.5)	94.9	
Molecular	Lumimal A	463 (41.3)	96.0	0.008
Subtype	Non-luminal A	657 (58.6)	82.1	

Abbreviations: EIC, extensive intraductal component; LVI, lymphovascular invasion; HG, histologic grade; LNR, lymph-node ratio; ECE, extracapsular extension.

**Table 4 cancers-11-00680-t004:** Multivariate analysis of disease-free survival.

Characteristics		Cox Regression Model		*p* Value
		Hazard Ratio	95% CI	
T stage	T1 vs. T2–4	2.628	1.489–4.638	0.001
LVI	(−) vs. (+)	1.92	1.071–3.441	0.028
HG	I or II vs. III	2.288	1.349–3.880	0.002
LNR	≤0.10 vs. >0.10	1.689	1.043–2.737	0.033
Molecular subtype	Luminal A vs. non-luminal A	1.029	0.695–2.121	0.496

Abbreviations: LVI, lymphovascular invasion; HG, histologic grade; LNR, lymph-node ratio.

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
