# Peer review of "Clinical Significance of Lymph-Node Ratio in Determining Supraclavicular Lymph-Node Radiation Therapy in pN1 Breast Cancer Patients Who Received Breast-Conserving Treatment (KROG 14-18): A Multicenter Study"

_cancers, 2019, doi:10.3390/cancers11050680_

Round 1
Reviewer 1 Report
The authors have conducted a thoughtful cohort analysis in patients treated with radiotherapy for pathologically node positive breast cancer. However, I do have some concerns regarding content that the reader may make and authors have made inferences from.
The claim that there is no difference between LNR > 0.1 and </= 0.1 in those patients receiving SCNRT is something I find misleading. Please include the number at risk on your plots or error margins on your estimates if you intend to keep figure 1. However, I think it would be more enlightening to the reader to reverse the subdivision. It is likely more useful to compare cumulative incidence of disease recurrence between patients receiving SCNRT and Non-SCNRT and using a competing risk model such as those proposed by Fine and Gray for your known covariates. Here you could focus on LNR > 0.10 and LNR </= 0.10 if you so choose.
It is not clear to me from the methodology whether the internal mammary lymph nodes were routinely irradiated. If not, the discussion should be expanded to explain the significance of this.
There are some minor typographical and grammatical errors that should be adjusted.
Author Response
Thank you for your valuable comments about our article.
We attach the file "the point to point response"
Thanks again for your thoughtful review.

Reviewer 2 Report
The aim of this retrospective study is to evaluate the clinical importance of lymph-node ratio for pN1 breast cancer patients. This study included large number of patients, and demonstrated some interesting points for readers.
Major points
1. This article’s constitution is strange. Introduction --> Result --> Discussion -> Material and Methods (common style: Introduction --> Material and Methods --> Result --> Discussion)
2. The median follow-up period was 60 months (in the abstract). BUT, the median follow-up time was 66 months (in the text). This points is a critical issue for scientific article.
3. Table 1 shows the patient’s characteristics, and there are some differences between non-SCNRT group and SCNRT group. The authors analyzed the clinical importance of lymph-node ratio using multi-variated analysis. But, they demonstrated that no difference of recurrence rate was found among SCNRT group. SCNRT group included relatively small number of patients, small proportion of LVI (-) which was associated with low-risk of loco-reginal recurrence, and small proportion of LNR. This analysis can’t demonstrate the clinical importance of LNR for SCNRT group. The authors should consider this analysis again.
Minor points
1. Page 3. LVI and HER2 need spell-out.
2. Page 4, line 98-101. Please check the grammar.
3. Page 6. The authors mentioned “… questions about the utility of SCNRT for early breast cancer patients have begun to emerge [27]”. But #27 was reported 10 years ago. They should cited other new references.
4. Page 7. Please clarify the ratio brachial plexopathy after conventional SCNRT of 50 Gy.
Author Response
Thank you for your valuable comments.
We attach the file "point to point response"
Thanks again for your thoughful review

Round 2
Reviewer 1 Report
The authors have addressed my comments.